# Bio-Multifunctional Sponges Containing Alginate/Chitosan/Sargassum Polysaccharides Promote the Healing of Full-Thickness Wounds

**DOI:** 10.3390/biom12111601

**Published:** 2022-10-31

**Authors:** Weiyan Quan, Puwang Li, Jinsong Wei, Yuwei Jiang, Yingye Liang, Weilin Zhang, Qizhou Chen, Kefeng Wu, Hui Luo, Qianqian Ouyang

**Affiliations:** 1The Marine Biomedical Research Institute, Guangdong Medical University, Zhanjiang 524023, China; 2South Subtropical Crop Research Institute, China Academy of Tropical Agricultural Sciences, Zhanjiang 524023, China; 3Orthopaedic Center, Affiliated Hospital of Guangdong Medical University, Zhanjiang 524001, China; 4The Marine Biomedical Research Institute of Guangdong Zhanjiang, Zhanjiang 524023, China

**Keywords:** *Sargassum pallidum* polysaccharides, bio-multifunctional, composite sponges, wound healing, alginate, chitosan

## Abstract

Creation of bio-multifunctional wound dressings with potent hemostatic, antibacterial, anti-inflammatory, and angiogenesis features for bolstering the healing of full-thickness wounds is sought after for clinical applications. We created bio-multifunctional composite sponges by coupling alginate and chitosan with *Sargassum pallidum* polysaccharides through electrostatic interactions, calcium ion (Ca^2+^) crosslinking, and lyophilization. Alginate/chitosan (AC) sponges with different concentrations of *Sargassum pallidum* polysaccharides were obtained and termed AC, ACS—1%, ACS—2.5%, and ACS—5%. ACS—1% and ACS—2.5% sponges exhibited uniform porosity, high water vapor transmission rate, high water absorption, as well as good hemostatic and antibacterial abilities. ACS—2.5% sponges facilitated wound closure and promoted angiogenesis and re-epithelialization in the dermis. These data suggest that ACS sponges containing a certain amount of *Sargassum pallidum* polysaccharides could be employed for treatment of full-thickness skin wounds.

## 1. Introduction

Skin is the main barrier against external environmental stimuli, including biological and chemical substances [1]. To maintain optimal physiological and biochemical conditions, the skin regulates body temperature, resists toxins/pathogens, secretes/excretes substances with antibacterial/bactericidal properties, and undertakes basic regulation of the immune system [2,3].

Skin wounds affect the health and life of humans [4]. Wound repair is a complicated and highly programmed process comprising hemostasis, inflammatory reactions, cell proliferation, and tissue remodeling [5]. Different approaches have been used to tackle the clinical aspects of wounding: hyperbaric oxygen therapy, negative-pressure therapy, growth factors, cell therapy, tissue-engineered skin, autograft transplantation, and “biological” dressings [6]. The latter are a novel kind of wound-dressing material based on “wet healing” theory. The good biocompatibility as well as the non-irritant and nontoxic properties of macromolecules have led to wound dressings based on natural polymers to be investigated [7]. For instance, marine polysaccharide-based biomaterials (which are inexpensive and found abundantly in many plants/animals), have garnered interest in wound healing (WH) applications [8]. In addition, alginate/chitosan composite materials can accelerate WH by bolstering coagulation, secretion of the extracellular matrix, formation of granulation tissue, antibacterial actions, and absorption of wound exudates [9,10,11]. Unfortunately, composite materials have difficulty in reducing scar formation, stimulating angiogenesis, and assisting the regeneration of hair follicles [12].

*Sargassum pallidum* is a type of perennial algae found along coastal regions of China and Japan. Polysaccharides from *S. pallidum* have antioxidant, anti-tumor, and immune-function properties [13]. Studies have shown that *S. pallidum* polysaccharides (SP) can activate the fibroblast growth factor/fibroblast growth factor receptor signal. The latter plays a part in the proliferation, migration, differentiation of different cell types and body development, as well as tissue/wound repair [14,15]. Previously, we revealed that SP has antioxidant and anti-apoptotic activities that can be harnessed to prevent skin-photoaging and to encourage skin health [16,17].

Along with biochemical features, the biophysical parameters of wound dressings can influence cell responses and WH. For instance, “sponge dressings” with porous structures have high porosity, strength, and specific surface area. These advantages can stop wound dehydration, speed-up tissue regeneration, as well as inhibit infection by stimulating macrophages to secrete cytokines [18,19]. Hence, scholars have focused energies towards sponge dressings that can improve the efficiency and quality of WH.

We created a series of composite sponges through electrostatic interactions, calcium ion (Ca^2+^) crosslinking, and freeze-drying. We assessed the physicochemical properties and effectiveness of composite sponges in a systematic fashion. We aimed to reveal new applications for SP to enable composite sponges to be employed as novel WH materials.

## 2. Materials and Methods

### 2.1. Ethical Approval of the Study Protocol

The study protocol was approved (SYXK (YUE) 2015–0147) by the ethics committee of Guangdong Medical University (Zhanjiang, China). Animals were handled in alignment with the Guidelines for the Care and Use of Laboratory Animals set by the Chinese National Institutes of Health (Beijing, China).

### 2.2. Materials

SP (molecular weight = 47,600 Da) is a water-soluble polysaccharide extracted from *S. pallidum*. It was identified and provided by Bo Rui Saccharide Biotechnology (Suzhou, China). Sodium alginate (viscosity = 200 mPa·s) was sourced from Shanghai Aladdin Bio-Chem Technology (Shanghai, China). Chitosan (molecular weight = 100 kDa; degree of deacetylation = 90%) was purchased from Qingdao Bozhi Huili Biotechnology (Qingdao, China). CaCl_2_, glycerol, and ethanol were of analytical grade. Institute of Cancer Research (ICR) mice were supplied by Guangdong Medical Laboratory Animal Center (Foshan, China).

### 2.3. Preparation of Alginate/Chitosan (AC) and Alginate/Chitosan/SP (ACS) Sponges

Alginate (3% *w*/*v*) was dissolved in deionized water (100 mL) with agitation for 1 h to create a homogeneous solution. Chitosan (3% *w*/*v*) was dissolved in 1% acetic-acid solution (50 mL) and added carefully to the alginate solution. The homogeneous gel solution was stirred at 500 rpm for 1–2 h at room temperature. This action was followed by freezing overnight at −20 °C and lyophilization in a freeze-dryer at −60 °C and 0.12 mbar under a vacuum for 48 h. The scaffolds thus formed were immersed in a solution of CaCl_2_/glycerol/ethanol mass (CaCl_2_):mass (glycerol):mass (ethanol) at a ratio of 7:3:90 for 6 h. Then, scaffolds were rinsed with an excess volume of deionized water, which were frozen at −20 °C for 24 h and then lyophilized at −80 °C (LGJ-10C, China) to obtain AC sponges.

Next, SP powder was dissolved in AC solution (as described above) followed by agitation for 1 h to elicit a homogenous solution. The SP solution and AC solution were mixed together by changing the weight ratio (i.e., *w*/*w*%) of SP to AC to create solutions of 1%, 2.5%, and 5%, which were denoted as ACS—1%, ACS—2.5%, and ACS—5%, respectively. Each ACS gel solution was transferred to a culture dish. The remainder of the procedure was followed as stated for the preparation of AC sponges. The schematic of the sponge synthesis method is shown in Figure 1.

### 2.4. Fourier-Transform Infrared (FTIR) Spectroscopy

FTIR spectroscopy of sponges (AC, ACS—1%, ACS—2.5%, ACS—5%) was undertaken using a Spectrum 100 (PekinElmer, Waltham, MA, USA) setup. Samples were prepared in potassium-bromide disks. FTIR spectroscopy was carried out at 4000–450 cm^−1^ at a resolution of 4 cm^−1^ over 16 scans.

### 2.5. Morphology of AC and ACS Sponges

AC sponges and ACS sponges were frozen in liquid nitrogen. Scanning electron microscopy (SEM) was undertaken using a S-4800 system (Hitachi, Tokyo, Japan). SEM enabled investigation of the surface structure and inner structure of AC sponges and ACS sponges.

### 2.6. Porosity of AC Sponges and ACS Sponges

A method based on liquid displacement was undertaken to ascertain the porosity of AC sponges and ACS sponges. Samples of sponges were immersed in absolute ethanol until they were saturated. The weight of sponge samples was documented before and after immersion in ethanol. The porosity (P) was calculated using the following equation:P = [(W2 − W1)/ρV] × 100%
where: W1 and W2 are the weight of samples before and after immersion in ethanol, respectively; V is the sample volume before immersion in ethanol; ρ is the density of ethanol.

### 2.7. Water Vapor Transmission Rate (WVTR)

Sponge samples were cut into circular segments of a 17 mm diameter. Physiologic (0.9%) saline (10 mL) was added to a 15-mL centrifuge tube. A sponge sample was attached to the mouth of the centrifuge tube, which was placed in an oven at 37 °C for 24 h. The centrifuge tube with 10 mL of Physiologic (0.9%) saline was directly placed in an oven at 37 °C for 24 h, which was the blank group.

Experiments were carried out in triplicate (at least). The WVTR was calculated using the following formula:WVTR = (M2 − M1)/πr^2^
where M1 and M2 was the weight of the centrifuge tube with 10 mL of Physiologic (0.9%) saline before and after which was placed in an oven at 37 °C for 24 h, and r denotes the radius of a circular sponge.

### 2.8. Water Absorption Ability

The water absorption ability of all the samples was tested. The water absorption capacity test was conducted according to the previous method [20]. AC and ACS sponges in dryness were accurately weighed and recorded as M0. The sponges were immersed in 37 °C water for 30 min, and the wet samples were put on the filter paper to absorb excess water. Then, the wet sponges were weighed and recorded as M1. Further, the water absorption ability of the sponges was calculated as:Water absorption (%) = (M1 − M0)/M0 × 100%

All experiments were done at least in triplicate.

### 2.9. In Vitro Blood-Clotting Study

Blood was collected from the ear vein of the New Zealand Rabbit. Blood was placed carefully into a tube containing 3.8% sodium citrate (ratio of blood:sodium citrate = 9:1 *v*/*v*). Blood clotting was tested according to the work of previous research [21]. Samples of AC sponges and ACS sponges were fashioned into circular shapes of ~20 mm diameter and placed on culture dishes. Next, blood (200 μL) was dropped onto the surface of each sample. Each sample was allowed to incubate for 5 min at 37 °C. Next, deionized water (50 mL) was poured slowly onto culture dishes from the edge. Red blood cells (RBCs) not trapped within blood clots would be hemolyzed in water. A solution containing hemoglobin was transferred to a cuvette to measure absorbance at 540 nm to ascertain the effect of composite sponges upon blood coagulation.

We wished to determine the influence of sponges upon hemocytes. Blood (0.5 mL) was added to the center of each sample to dampen the sponge. After 2 min, each sponge was fixed with 2.5% glutaraldehyde, and excess blood washed away with 0.9% saline. The sponge was washed (15 min each) with a gradient of alcohol solutions (75%, 85%, 95%, 100%). After drying and coating with gold, the sponge underwent imaging by SEM.

### 2.10. Cytotoxicity

Investigation of cytotoxicity was done using human umbilical vein endothelial cells (HUVECs using Cell Counting Kit-8 (CCK—8, purchased from Beyotime Biotechnology). The cells were donated to Kunming Institute of Zoology, Chinese Academy of Sciences. A cell suspension (5 × 10^4^ cells/mL) was seeded into a 96-well plate (100 μL cells/well). Cells were mixed rigorously before seeding to ensure an equal density of cells in each well. Upon cell attachment, 10 μL of the sample extract (0, 0.5, 1.0, 2.0, 4.0, 6.0 mg/mL) was mixed Dulbecco’s modified Eagle’s medium (90 μL) containing 10% (*v*/*v*) fetal bovine serum and antibiotics (1% *v*/*v*). The group with 0 mg/mL of the sample extract was the control group. Next, HaCaT cells were allowed to incubate with the mixture in a 96-well plate for 24 h. The cell number was ascertained using the CCK-8 assay. The cell viability of each sample was determined using the following equation:Cell viability = Absorbance of sample at 490 nm/Absorbance of the control × 100%

### 2.11. Antibacterial Activity

The antibacterial activity of sponges against Escherichia coli (Gram—negative) and Staphylococcus aureus (Gram—positive) was measured according to the work of Ouyang and colleagues [21]. Samples of sponges were immersed in a bacterial suspension (2 mL) for 24 h at 37 °C. The bacterial suspension was diluted to 6 × 10^4^. An aliquot (15 µL) was spread onto sponges. The number of bacterial colonies formed was counted (*n* = 3). The bacterial viability of samples was calculated using the following formula:Bacterial viability = (Number sample)/(Number control) × 100%

### 2.12. WH In Vivo

WH behavior was determined by covering dressing materials onto the injured skin of mice. Mice (30 ± 5 g) were divided randomly into four groups. After the induction of anesthesia (1% pentobarbital sodium for 15 min), the fur on mice backs was shaved and disinfected with 70% alcohol. A circular full-thickness wound (~15 mm) was created on the back. Control-group mice had their wound covered with gauze as a treatment. AC sponges and ACS sponges were applied to the wound. Spongy dressings were changed on a daily basis. Mice were killed at different times. Images were taken. The area of the wound closure (S) was calculated using this equation:S = [(S0 − S1)/S0] × 100%
where S0 denotes the wound area and S1 represents the unhealed wound area.

### 2.13. Histology

Upon killing the ICR mice at specific time points, the skin and muscle around the wound area were harvested, and fixed in 4% paraformaldehyde. Tissue samples were stained with hematoxylin and eosin (H&E) and observed under a fluorescence microscope (IX73; Olympus, Tokyo, Japan). Tissue samples on day-21 were stained with Masson’s trichrome to measure total collagen content (calculated as a percentage of aniline-blue staining in the dermis). Slides were photographed with a digital slice scanning system (Pannoramic MIDI; 3D HISTCH, Budapest, Hungary) and quantified by Image-Pro Plus 6.0 (Media Cybernetics, Rockville, MD, USA).

### 2.14. Statistical Analyses

Results were analyzed with SPSS 22 (IBM, Armonk, NY, USA) using independent-sample t-tests. Numerical data are given as the mean ± SD. Differences among groups were analyzed using one-way ANOVA. *p* < 0.05 was considered significant.

## 3. Results

### 3.1. Preparation and Characterization of AC Sponges and ACS Sponges

AC solutions with different amounts of SP were used to create sponges (ACS—1%, ACS—2.5%, ACS—5%). Composite sponges were fashioned as wound dressings through electrostatic interaction, Ca^2+^ crosslinking, and freeze-drying. The negatively charged AG, which possesses the ability to form strong electrostatic interaction with cationic, can interact with the cationic charged amine group of the CS unit to form a polyelectrolyte mixture. The addition of SP into the AC solution may make more complex ionic interactions possible.

The structure of AC sponges and ACS sponges was evaluated by FTIR spectroscopy. CS is the only naturally occurring alkaline polysaccharide with a positive charge. CS can interact with negatively charged Alginate to form a “polyelectrolyte mixture”. The addition of negatively charged SP into the AC solution enables complex ionic interactions to occur.

We wished to identify the functional groups and molecular interactions in composite sponges. Hence, FTIR spectroscopy of marine polysaccharides (and their composite sponges) was carried out.

Obvious absorption peaks on the AC spectral line at 1606 cm^−1^ and 1417 cm^−1^ were noted (Figure 2A), which could be attributed to the asymmetric and symmetric stretching vibrations of carboxylic groups, respectively. A peak at 1037 cm^−1^ could be ascribed to a C–OH stretching vibration [22].

The absorption band of CS between 3000 cm^−1^ and 3500 cm^−1^ was here determined by the overlapping of the –OH stretch and –NH stretch vibrations. The bands at 2881 and 1656 cm^−1^ were the absorption of the –CH stretch vibration and amide group I, respectively. While the band at 1596 cm^−1^ was determined by the –NH deformation wave, and the band at 1424 cm^−1^ was the absorption band of –CH_2_ bending and –CH_3_ deformation [23].

With respect to the AC composite sponge, the peak intensity at 1629 cm^−1^ could be attributed to the superposition of the carboxylate group of AG and amine group of CS. This peak suggested that the –NH_2_ group of CS in acidic conditions was altered to –NH_3_^+^ and interacted in an electrostatic manner with the –COO^−^ group of AG [13]. Similarly, the absorption peak became lower near 1037 cm^−1^, possibly because of the interaction between the –C–OH and NH_3_^+^ group on the sponge surface. A higher stretching frequency in –OH was documented from 3326 cm^−1^ to 3411 cm^−1^, thereby indicating the presence of intermolecular hydrogen bonds in the AG–CS system [24].

The FTIR spectrum of SP revealed characteristic peaks at 2910 cm^−1^, 1647 cm^−1^, and 1420 cm^−1^, which could be ascribed to C–H, C=O, and C–N stretching vibration peaks. Peaks at 1200–1000 cm^−1^ and 860 cm^−1^ could be attributed to the characteristic peaks of SP [24]. The FTIR spectra of ACS sponges included all of the characteristic peaks present in AC and SP. Hence, composite sponges were created through electrostatic interactions (Figure 2B).

The surface characteristics of AC sponges and ACS sponges were documented at macroscopic and microscopic levels. SP had a homogeneous distribution in ACS sponges (Figure 2C). AC, ACS—1%, and ACS—2.5% sponges had a uniform porous structure within their interior cavities—a three-dimensional and interconnected pore structure with an irregular arrangement. The pore structure and porosity of sponges have been reported to be crucial parameters for wound healing that can influence the absorption of exudates and blood [21]. The addition of 5% SP had significant effects on the formation and morphology of sponges because their scaffolds collapsed. Scanning electron micrographs showed that the quality of composite sponges was highly dependent upon the weight ratio of SP (i.e., a higher mass ratio of SP led to an inferior quality of sponges). We discovered that ACS—1% sponges and ACS—2.5% sponges had uniform porosity.

### 3.2. Porosity, WVTR and Water Absorbability Capacity of AC Sponges and ACS Sponges

High porosity helps wound dressings to absorb exudates from the injury surface and initiate gas exchange [21]. The porosity of AC sponges and ACS sponges was evaluated using the ethanol-displacement method (Figure 3A). The porosity of AC sponges was 52%. SP addition to AC sponges led the latter to have a higher porosity than AC sponges. As the SP content increased, the porosity of ACS—1% and ACS—2.5% composite sponges increased slowly. Otherwise, the porosity of ACS—5% sponges was lower than that of ACS—1% and ACS—2.5% sponges, but significantly higher than that of AC sponges. This phenomenon could be because addition of excess SP to AC sponges led to scaffold collapse and non-uniform porosity.

An appropriate WVTR (water vapor transmission rate) is crucial for assisting WH (wound healing) [25,26]. A higher WVTR leads to faster scar formation. A low WVTR leads to accumulation of exudates and increased bacterial growth. A wound dressing with an “ideal” WVTR would prevent water loss from a wound, and prevent exudate formation in the wound bed, which can exchange gas with the external environment to maintain a microenvironment in the wound that can promote wound healing [27,28]. The WVTR of AC, ACS—1%, and ACS—2.5% sponges was not significantly different from that of the blank group, but the WVTR in the ACS—5% group was significantly lower than that of the blank group (Figure 3B). These data suggested that ACS—1% and ACS—2.5% sponges were appropriate for the healing of skin wounds.

Water absorption and retention are essential wound dressing qualities that can be utilized to assess the effectiveness of wound dressings in removing wound exudation and maintaining a moist environment around the wound. Meanwhile, sponges with high capacity can concentrate blood cells and accelerate blood coagulation. As seen from Figure 2C, AC and ACS sponges possessed a great deal of water absorption. Among the samples, ACS—2.5% has the highest capacity, which was appropriate for the healing of skin wounds (Figure 3C).

### 3.3. Hemostatic Performance of AC Sponges and ACS Sponges

Extensive full-thickness wounds can lead to massive bleeding. Hence, an efficient hemostatic performance of a wound dressing is crucial for accelerating WH [12]. In vitro experiments based on whole-blood coagulation were carried out to evaluate the hemostatic ability of composite sponges.

A large difference was noted in the adsorption of hemocytes among AC sponges and ACS sponges. The latter had good absorptivity, and RBCs were absorbed on pores, which led to potent coagulation (Figure 4C). The ACS sponge–RBC interaction was strong, and some RBCs were adsorbed. For deeper comprehension of the hemostatic performance of composite sponges, hemoglobin concentrations in samples were obtained based on their absorbance values. A higher absorbance value of a hemoglobin solution suggested less blood coagulation. Absorbance values on ACS sponges were significantly lower than those on AC sponges (Figure 4B). ACS—2.5% sponges had the lowest absorbance values, which may suggest a better repairing effect.

### 3.4. Biocompatibility and Antibacterial Ability of AC Sponges and ACS Sponges

Sponges with good biocompatibility are crucial for biomedical applications. The cytocompatibility of composite sponges was examined using the CCK-8 assay. HUVECs belonged to endothelial cells, located on the inner surface of blood vessels, were the main cellular components involved in neovascularization. After skin injury, decreased blood supply and accelerated cell metabolism lead to hypoxia at the injured site. Tissue hypoxia induces high expression of HIF1 in a variety of cells, stimulated the secretion of pro-angiogenic factors, and finally activated the proliferation and migration of endothelial cells to promote neovascularization [29]. Natural AG and natural CS have outstanding biocompatibility in biomedical applications [30,31,32,33]. AC sponges and ACS sponges had good cytocompatibility (Figure 5A). SP addition to composite sponges can promote the proliferation activity of HUVECs. Thus, ACS composite sponges could be employed safely as wound-dressing materials.

In tandem with preventing wound infection from bacteria, an “ideal wound” dressing will have inherent antibacterial features [21]. The antibacterial activity of composite sponges against *E. coli* and *S. aureus* was studied in vitro. AC sponges and ACS sponges had excellent properties against *S. aureus* (Figure 5B). All samples exhibited lower activity against *E. coli* than *S. aureus*, because the structure of the cytoderm of *E. coli* is more complicated than that of the *S. aureus*, while ACS—2.5% sponges had optimal properties against *E. coli* (Figure 5C). The excellent antibacterial activity of composite sponges showed potential to efficiently prevent the wound from the bacteria infection. The cationic charged amine groups of CS can interact with negatively charged bacteria and then penetrate the walls of the bacteria. The outstanding antibacterial activity of ACS—2.5% sponges could be due to the combined antibacterial activity of chitosan, sodium alginate, and SP. Therefore, ACS—2.5% sponges could be attractive for WH applications.

### 3.5. Healing of Full-Thickness Skin Wounds In Vivo

We wished to assess the WH capacity of AC sponges and ACS sponges in vivo. The full-thickness wounds of ICR mice treated with composite sponges. The wounds treated with AC, ACS—1%, ACS—2.5%, ACS—5% sponges and sterile gauze (control) were observed for 7, 14, and 21 days after treatment, respectively (Figure 6). Qualitative analyses from photographs indicated that the wound areas in all samples shrank as the treatment duration increased. On day-7, the scab was on the wound bed in all samples; except for the ACS—5% group and AC group, the wound area in the ACS—1% and ACS—2.5% group was significantly smaller than that in the control group. On day-14, peripheral scabs began to fall-off, and the area continued to decrease in size. Wound-area reductions were greater in the AC, ACS—1%, and ACS—2.5% groups than in the control group and ACS—5% group. Optimal outcomes were seen in the ACS—1% group, where the scab had fallen-off completely.

Alterations in the healing rates of wounds in ICR mice are shown in Figure 7. On day-3, the skin wounds in each group started to form scabs. On day-7, the scabs began to shrink, with the slowest healing rate being seen in the control group. The healing rate of ACS—1% and ACS—2.5% groups was significantly faster than that of the control group, and they had a faster healing rate than that of the other groups. The changes observed on days 14 and 21 were similar to those observed on day-7, and the healing rate of ACS—1% and ACS—2.5% groups was >95%, which was significantly faster than that of AC sponges and ACS—5% sponges. These results suggested that ACS—1% sponges and ACS—2.5% sponges accelerated WH compared with other sponges thanks to wound contraction, which was attributed to SP addition to sponges. However, the repairing effect of AC sponges upon addition of 5% SP had no significant effect compared with that using AC sponges because addition of excess SP affected the structure of AC sponges, which led to poor hemostatic (Figure 4) and antibacterial (Figure 5B,C) performances.

### 3.6. Histomorphology

WH involves hemostasis, inflammation, cell proliferation/migration, angiogenesis, and tissue remodeling [34]. We wished to assess the growth rate and quality of regenerated tissues. Hence, histopathological analyses were undertaken on H&E-stained tissues.

Formation of thick epidermal and granulation tissue are crucial features when assessing WH in the proliferative phase [34]. Epidermal thickness reflects the WH rate accompanied by scar formation. In other words, the thickness of epidermis reflects the rate of wound healing companied by scar formation. Seen from Figure 8, on day-14, ACS sponges formed thicker epidermis. New blood vessels are essential to wound healing by delivering oxygen and nutrients to support cellular activity and regulate the damaged tissue microenvironment. However, mild inflammation and potent angiogenesis were noted in the AC group and ACS—1% group. Complete regeneration of the epidermis was observed in the ACS—2.5% group, which indicated that this formulation bolstered WH. On day-21, more blood vessels were observed in the ACS—2.5% group, which suggested that the repairing effect was proceeding apace.

In general, the tamponade effect offered by the porous structure and SP in ACS—2.5% sponges aided the reconstruction of full-thickness skin wounds.

## 4. Conclusions

Bio-multifunctional ACS composite sponges were developed through electrostatic interactions and Ca^2+^ crosslinking. ACS—1% and ACS—2.5% sponges exhibited uniform porosity and a high WVTR, which aided the healing of skin wounds. ACS—2.5% sponges had more potent hemostatic and antibacterial abilities in vitro and aided wound closure compared with AC, ACS—1%, and ACS—5% sponges in a model of full-thickness wounds in ICR mice. Compared with other groups, more blood vessels were observed in the ACS—2.5% group, which bolstered dermal re-epithelialization significantly. Therefore, ACS—2.5% sponges have great potential as dressings to promote WH.

## Figures and Tables

**Figure 1 biomolecules-12-01601-f001:**
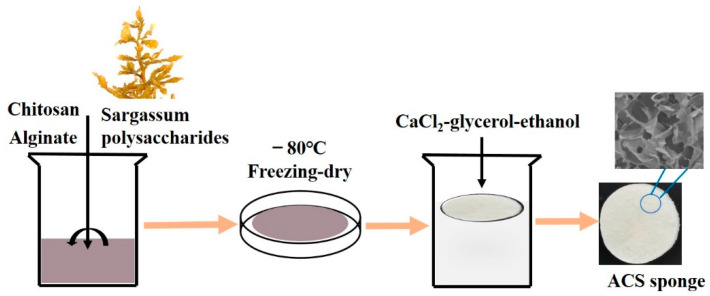
The schematic of the sponge synthesis method.

**Figure 2 biomolecules-12-01601-f002:**
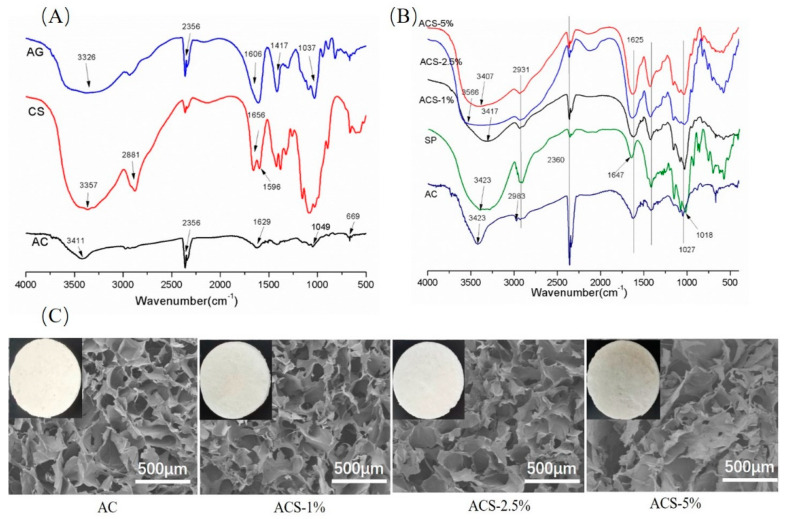
(**A**) FTIR spectra of AG, CS, and AC. (**B**) FTIR spectra of AC, SP, ACS—1%, ACS—2.5%, and ACS—5%. (**C**) Scanning electron micrographs of composite sponges; the insets in images are photographs of the corresponding composite sponges.

**Figure 3 biomolecules-12-01601-f003:**
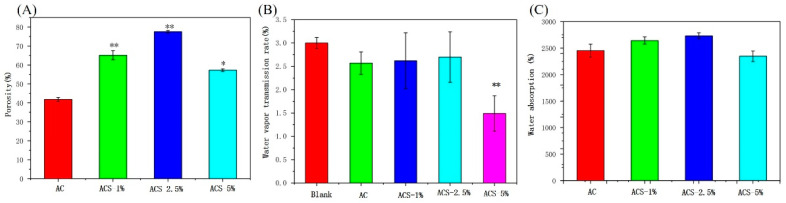
(**A**) Porosity of AC, ACS—1%, ACS—2.5%, and ACS—5% sponges. Data are the mean ± SD for each group (n = 3), * Significant difference from the AC group, * *p* < 0.05, ** *p* < 0.01. (**B**) Water vapor transmission rate of blank, AC, ACS—1%, ACS—2.5%, and ACS—5% sponges. Data are the mean ± SD for each group (n = 3), * Significant difference from the blank, ** *p* < 0.01. (**C**) Water absorption capacity of AC, ACS—1%, ACS—2.5%, and ACS—5% sponges.

**Figure 4 biomolecules-12-01601-f004:**
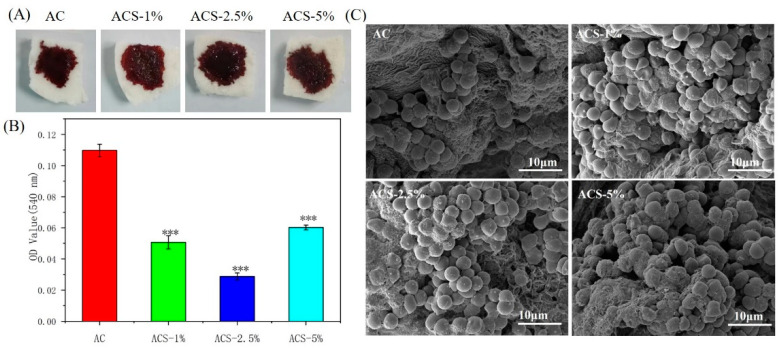
In vitro blood-coagulation test of composite sponges. (**A**) Photographs of the blood-clotting process. (**B**) Hemoglobin concentration in water. Data are the mean ± SD for each group (n = 3), * Significant difference from the AC group, *** *p* < 0.001. (**C**) Scanning electron micrographs of three sponges after coagulation.

**Figure 5 biomolecules-12-01601-f005:**
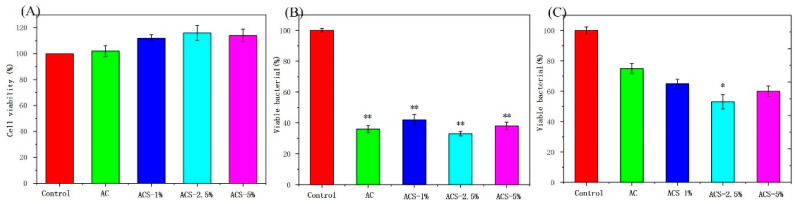
(**A**) Cytotoxicity testing. (**B**) Survival of Staphylococcus aureus after contact with different sponges for 24 h. (**C**) Survival of Escherichia coli after contact with different sponges for 24 h. Data are the mean ± SD for each group (*n* = 3), * Significant difference from the control group, * *p* < 0.05, ** *p* < 0.01.

**Figure 6 biomolecules-12-01601-f006:**
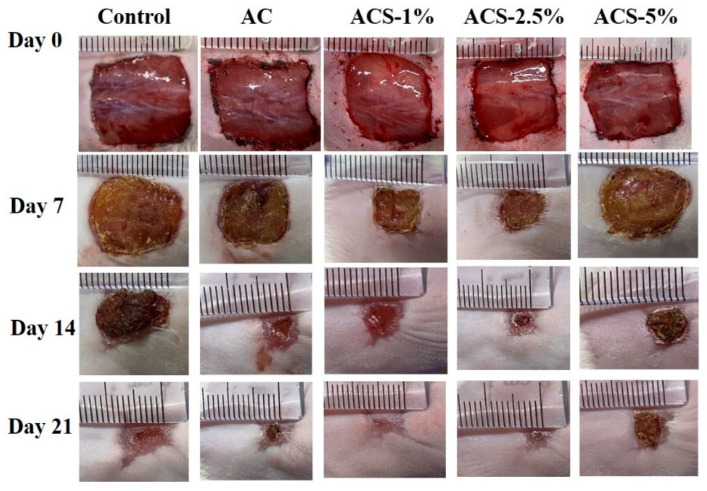
Photographs of wounds treated by sterile gauze (control), AC, ACS—1%, ACS—2.5%, and ACS—5% sponges at 0, 7, 14, and 21 days.

**Figure 7 biomolecules-12-01601-f007:**
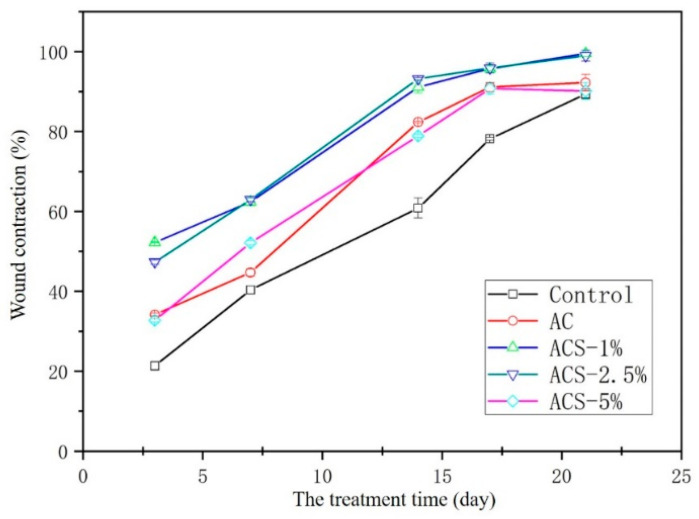
Wound contraction for treatment by sterile gauze (control), AC, ACS—1%, ACS—2.5%, or ACS—5% sponges.

**Figure 8 biomolecules-12-01601-f008:**
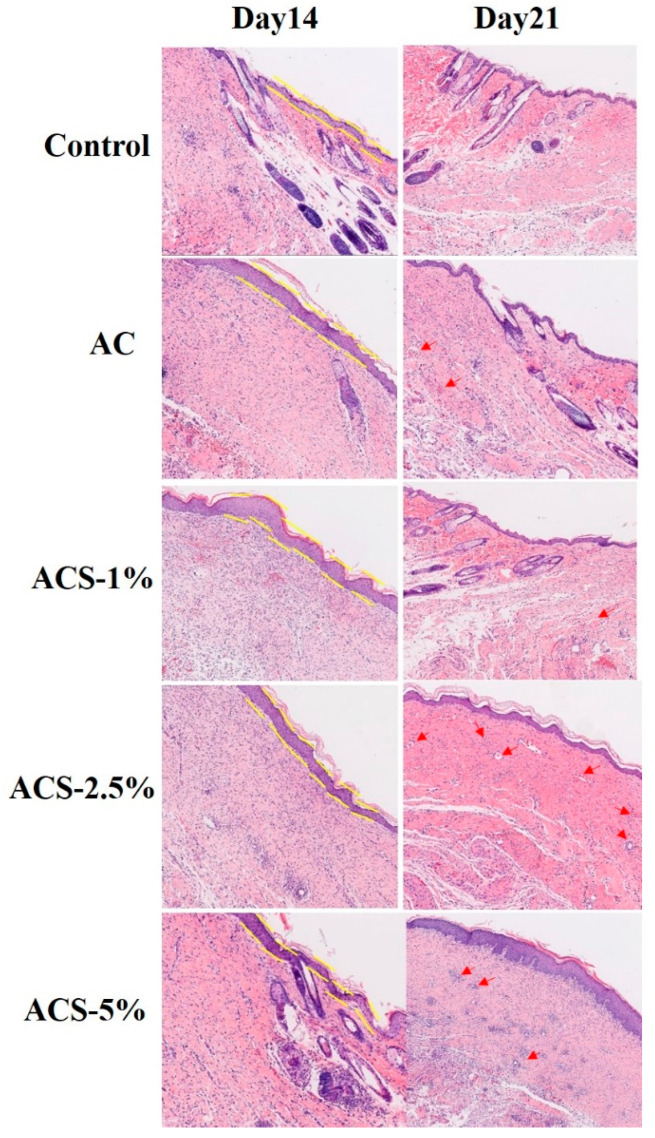
Regenerated skin tissues in wound sites after treatment with composite sponges at 14 days and 21 days as detected by H&E staining (epidermal thickness: yellow lines; blood vessels: red arrows).

## Data Availability

The data presented in this study are available on request from the corresponding author.

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
