# Peer review of "Bio-Multifunctional Sponges Containing Alginate/Chitosan/Sargassum Polysaccharides Promote the Healing of Full-Thickness Wounds"

_biomolecules, 2022, doi:10.3390/biom12111601_

Round 1

Reviewer 1 Report

For this research article, the authors have selected potential biomaterials focusing their wound healing capability.  Manuscript is well written encompassing sufficient testing.  Results need to be adequately discussed. There are few gaps which have to be addressed.

1.      Properly discuss the electrostatic interactions and calcium  ion (Ca2+) crosslinking while preparation of AC and ACS sponges

2.      Lines 232-234.  It is to clarify these peaks in FTIR are for AG (Alginate) or AC. As per the Figure 1, peaks described in lines 232-235 correspond to AG. Moreover characteristic peaks of Chitosan should be described separately.

3.      Line no. 254 replace PS with SP.

4.      Line 176. Clearly describe control.

Author Response

Reviewer #1

For this research article, the authors have selected potential biomaterials focusing their wound healing capability.  Manuscript is well written encompassing sufficient testing. Results need to be adequately discussed.

  1. Properly discuss the electrostatic interactions and calciumion (Ca2+) crosslinking while preparation of AC and ACS sponges

Response: Thank you for your kind suggestion, we have had a discussion about the electrostatic interactions and calcumion (Ca2+) crosslinking while preparation of AC and ACS sponges on page 5, line 222-226. The negatively charged AG which possesses the ability to form strong electrostatic inter-action with cationic, can interact with the cationic charged amine group of the CS unit to form a polyelectrolyte mixture. The addition of SP into the AC solution may make more complex ionic interactions possible.

  1. Lines 232-234. It is to clarify these peaks in FTIR are for AG (Alginate) or AC. As per the Figure 1, peaks described in lines 232-235 correspond to AG. Moreover characteristic peaks of Chitosan should be described separately.

Response: Thank you for your kind suggestion, we have described the characteristic peaks of Chitosan on page5, line243-247. The absorption band of CS between 3000 cm−1 and 3500 cm−1 was here determined by the overlapping of -OH stretch and -NH stretch vibrations. The bands at 2881 and 1656 cm−1 were the absorption of -CH stretch vibration and amide group I, respectively. While the band at 1596cm−1 was determined by -NH deformation wave, and the band at 1424 cm−1 was the absorption band of -CH2 bending and -CH3 deformation [23].

[23] Brugnerotto, J.; Lizardi, J.; Goycoolea, F.M.; Argüelles-Monal, W.; Desbrières, J.; Rinaudo, M. An infrared investigation in relation with chitin and chitosan characterization. Polymer., 2001,42 (8), 3569–3580.

  1. Line no. 254 replace PS with SP.

Response: Thank you for your kind reminder. We have corrected it on page 6, line 266.

  1. Line 176. Clearly describe control.

Response: Thank you for your kind reminder. Upon cell attachment, 10 μL of the sample extract (0, 0.5, 1.0, 2.0, 4.0, 6.0 mg/mL) was mixed Dulbecco’s modified Eagle’s medium (90 μL) containing 10% (v/v) fetal bovine se-rum and antibiotics (1% v/v). The group with 0 mg/mL of the sample extract was the control group.

Reviewer 2 Report

Weiyan et al. investigated the generation of a bio-multifunctional wound dressing to bolster wound healing. They state that Alginate/chitosan sponge with 2.5% Sargassum pallidum polysaccharides have great potential as dressing to promote wound healing. Although the data is promising, and ACS-2.5% sponges may be a great option for healing, the manuscript requires major changes. The manuscript suffers from a poorly written methods section which is missing vital information, a speculative result section that needs additional analysis, and an incomplete conclusion. A better explanation of the reasoning for the experimental design is needed, along with better conclusive statements that capture the findings and correlates them to outside literature. 

Points needed to be addressed;

-A more informative methods section is needed, for example, how the sponges were lyophilized (lines 99 and 100) and how many samples were used in each experiment. 

-Results section has multiple speculative statements. Such as, in line 274, why should it be assumed that there would be a smaller pore size when Figure 1C does not show this? Is there a way to quantitively measure the pore size? Speculative statements should be removed from the results section and focus only on the data that is obtained. 

-Too many acronyms in the paper; please write out WH and WVTR in Line 280. Please define the blank group in line 287; the significance is not clear for the water vapor transmission rate. A better explanation of the reasoning for the experimental design is needed. 

-More descriptive figure legends are needed, such as the duration of time, number of samples, and a summary sentence of the finding. Bar graphs are not sufficient to demonstrate the data; bar graphs with individual data points can better represent the findings. 

-The result section is poorly written and missing a proper explanation of methods and reasoning. An example is on lines 312-314, stating that "Human 312 umbilical vein endothelial cells (HUVECs) were chosen because they play an essential part in WH" is insufficient. HUVECs are found in development; how would these cells play an essential part in WH? Authors should refrain from making strong statements such as in line 316-317 "SP addition to composite sponges boosted the proliferative activity of HU-VECs considerably" the graph in figure 4A does not show a significant difference. Histology of the seeded scaffold and immunohistochemistry staining (for PCNA and BAX) are needed to understand if this change can be visualized. A better and clearer explanation is needed in the results regarding the description of antibacterial features.

-In Figure 5, please only show the scar area at Day 0 like the other Days, a full image of the mouse is unnecessary. 

-Figure analysis is necessary for Figure 7. The image does not support statements in the results. "On day-14, a moderate inflammatory reaction was present; epithelial regeneration could not be seen in the control group and ACS-5% group. However, mild inflammation and potent angiogenesis ….." Additional stainings and higher magnification images with arrows indicating what the authors state are necessary. Simple H&E staining at a low magnification does not support these statements.

Author Response

Weiyan et al. investigated the generation of a bio-multifunctional wound dressing to bolster wound healing. They state that Alginate/chitosan sponge with 2.5% Sargassum pallidum polysaccharides have great potential as dressing to promote wound healing. Although the data is promising, and ACS-2.5% sponges may be a great option for healing, the manuscript requires major changes. The manuscript suffers from a poorly written methods section which is missing vital information, a speculative result section that needs additional analysis, and an incomplete conclusion. A better explanation of the reasoning for the experimental design is needed, along with better conclusive statements that capture the findings and correlates them to outside literature.

Points needed to be addressed;

  1. A more informative methods section is needed, for example, how the sponges were lyophilized (lines 99 and 100) and how many samples were used in each experiment.

Response: Thank you for your kind suggestion, we have added the method of lyophilization. The scaffolds were rinsed with an excess volume of deionized water, which were frozen at -20℃ for 24 h and then lyophilized at -80℃(LGJ-10C, China) to obtain AC sponges. The amount of sample used for each experiment was described in each test method.

  1. Results section has multiple speculative statements. Such as, in line 274, why should it be assumed that there would be a smaller pore size when Figure 1C does not show this? Is there a way to quantitively measure the pore size? Speculative statements should be removed from the results section and focus only on the data that is obtained.

Response: Thank you for your kind suggestion. We have removed the speculative statements.

  1. Too many acronyms in the paper; please write out WH and WVTR in Line 280. Please define the blank group in line 287; the significance is not clear for the water vapor transmission rate. A better explanation of the reasoning for the experimental design is needed.

Response: Thank you for your kind suggestion. WH represents wound healing and WVTR represents water vapor transmission rate. We have written out in line 292-293. The centrifuge tube with 10 mL of Physiologic (0.9%) saline was directly placed in an oven at 37℃ for 24 h, which was the blank group. We have defined the blank group in line139-148. And we have explained the reasoning for the needed experimental design in line294-299. According to the research, the dressing with proper WVTR can exchange gas with the external environment to maintain a microenvironment in the wound that can promote wound healing, which can control water loss from the wound site and was essential for an ideal wound dressing. The dressing with good air permeability can avoid the propagation of anaerobic bacteria and the accumulation of tissue exudate, and reduce the occurrence of wound inflammation.

  1. More descriptive figure legends are needed, such as the duration of time, number of samples, and a summary sentence of the finding. Bar graphs are not sufficient to demonstrate the data; bar graphs with individual data points can better represent the findings.

Response: Thank you for your kind suggestion. We have changed some of the figures in the manuscript.

  1. The result section is poorly written and missing a proper explanation of methods and reasoning. An example is on lines 312-314, stating that "Human 312 umbilical vein endothelial cells (HUVECs) were chosen because they play an essential part in WH" is insufficient. HUVECs are found in development; how would these cells play an essential part in WH? Authors should refrain from making strong statements such as in line 316-317 "SP addition to composite sponges boosted the proliferative activity of HU-VECs considerably" the graph in figure 4A does not show a significant difference. Histology of the seeded scaffold and immunohistochemistry staining (for PCNA and BAX) are needed to understand if this change can be visualized. A better and clearer explanation is needed in the results regarding the description of antibacterial features.

Response: Thank you for your kind suggestion. HUVECs were chosen in this paper. The reason was that HUVECs belonged to endothelial cells, located on the inner surface of blood vessels, were the main cellular components involved in neovascularization. After skin injury, decreased blood supply and accelerated cell metabolism lead to hypoxia at the injured site. Tissue hypoxia induces high expression of HIF1 in a variety of cells, stimulated the secretion of pro-angiogenic factors, and finally activated the proliferation and migration of endothelial cells to promote neovascularization. We have modified the statement. The results regarding the description of antibacterial features have been modified to make a better and explanation.

  1. In Figure 5, please only show the scar area at Day 0 like the other Days, a full image of the mouse is unnecessary.

 Response: Thank you for your kind suggestion. We have changed the figure in line 372.

  1. Figure analysis is necessary for Figure 7. The image does not support statements in the results. "On day-14, a moderate inflammatory reaction was present; epithelial regeneration could not be seen in the control group and ACS-5% group. However, mild inflammation and potent angiogenesis ….." Additional stainings and higher magnification images with arrows indicating what the authors state are necessary. Simple H&E staining at a low magnification does not support these statements.

Response: Thank you for your kind suggestion. We have modified Figure7. The yellow lines represented epidermal thickness, and red arrws represented blood vessels.  

Reviewer 3 Report

In this study, a multifunctional sponge containing alginate, chitosan, and sargassum polysaccarides was synthesized. The results show that the ACS sponge with a certain amount of Sargassum pallidum can be a good candidate to be employed for the treatment of full-thickness skin wounds. In my opinion, the paper can be published, possibly only if the following issues are addressed.

  1. The abstract and introduction need to be modified to better present the novelty of this research.
  2. The style and color of the graphs could be further beautified.
  3. Adding a schematic figure for the sponge synthesis method can help to better understand the synthesis method.
  4. The SEM image is not enough to evidence the influence of Sargassum pallidum on the pore structure and porosity of sponges. More SEM images are needed on smaller scales.
  5. Mechanical properties are an important aspect of sponge wound dressing. An ideal wound dressing has the proper mechanical properties. What about the mechanical properties of the prepared sponges? What is the effect of adding Sargassum pallidum to alginate/chitosan?
  6. Water absorption and retention are essential wound dressing qualities that can be utilized to assess the effectiveness of wound dressings in removing wound exudation and maintaining a moist environment around the wound. What is the water absorption and water retention of prepared sponges? What is the effect of adding Sargassum pallidum on these properties of the sponges?

Author Response

In this study, a multifunctional sponge containing alginate, chitosan, and sargassum polysaccarides was synthesized. The results show that the ACS sponge with a certain amount of Sargassum pallidum can be a good candidate to be employed for the treatment of full-thickness skin wounds. In my opinion, the paper can be published, possibly only if the following issues are addressed.

  1. The abstract and introduction need to be modified to better present the novelty of this research.

Response: Thank you for your kind suggestion. The abstract and introduction have been modified, which was marked in red in the paper.

  1. The style and color of the graphs could be further beautified.

Response: Thank you for your kind suggestion. We have changed the style and color of some figures to make them further beautified.

  1. Adding a schematic figure for the sponge synthesis method can help to better understand the synthesis method.

Response: Thank you for your kind suggestion. We have added the schematic figure for the sponge synthesis method in Figure in line 111-112.

  1. The SEM image is not enough to evidence the influence of Sargassum pallidum on the pore structure and porosity of sponges. More SEM images are needed on smaller scales.

Response: Thank you for your kind suggestion. I agree with your opinion. Therefore, we have evaluated the porosity of AC sponges and ACS sponges, and the results in Figure 2A showed that the porosity of AC sponges was 52%. SP addition to AC sponges led the latter to have a higher porosity than AC sponges. As the SP content increased, the porosity of ACS-1% and ACS-2.5% composite sponges in-creased slowly. Otherwise, the porosity of ACS-5% sponges was lower than that of ACS-1% and ACS-2.5% sponges, but significantly higher than that of AC sponges. This phenome-non could be because addition of excess SP to AC sponges led to scaffold collapse and non-uniform porosity (seen in Figure1C).

  1. Mechanical properties are an important aspect of sponge wound dressing. An ideal wound dressing has the proper mechanical properties. What about the mechanical properties of the prepared sponges? What is the effect of adding Sargassum pallidum to alginate/chitosan?

Response: Thank you for your kind suggestion. The composite sponges have certain mechanical properties with the electrostatic interactions and calcumion (Ca2+) crosslinking. When the sponges were compressed and bent, ACS sponges also exhibited their flexible mechanical properties. It indicated that the addition of SP did not affect the mechanical properties of AC sponge.

  1. Water absorption and retention are essential wound dressing qualities that can be utilized to assess the effectiveness of wound dressings in removing wound exudation and maintaining a moist environment around the wound. What is the water absorption and water retention of prepared sponges? What is the effect of adding Sargassum pallidum on these properties of the sponges?

Response: Thank you for your kind suggestion. Water absorption and retention are essential wound dressing qualities that can be utilized to assess the effectiveness of wound dressings in removing wound exudation and maintaining a moist environment around the wound. We have supplemented the prepared sponges with experiments on its ability to absorb water. As seen from Figure2C, AC and ACS sponges possessed a great deal of water absorption. Among the samples, ACS-2.5% has the highest capacity, which was appropriate for the healing of skin wounds.  

Sargassum pallidum is a type of perennial algae found along coastal regions of China and Japan. Polysaccharides from S. pallidum have antioxidant, anti-tumor, and immune-function properties. Studies have shown that S. pallidum polysaccharides (SP) can activate the fibroblast growth factor/fibroblast growth factor receptor signal. The latter plays a part in the proliferation, migration, differentiation of different cell types and body development, as well as tissue/wound repair. Previously, we revealed that SP had antioxidant and anti-apoptotic activities that can be harnessed to prevent skin-photoaging and to encourage skin health. Therefore, we hypothesized that the addition of Sargassum polysaccharide could synergistically promote wound repair. The results showed that ACS-2.5% sponges have great potential as dressings to promote wound healing.

Round 2

Reviewer 2 Report

The authors have sufficiently addressed the concerns. 

Reviewer 3 Report

The authors corrected the manuscript in accordance with the suggestions of the reviewer.